# Identification of sources of male sterility in the Colombian Coffee Collection for the genetic improvement of *Coffea arabica* L.

**Juan Carlos Arias Suárez** *[ID]*\*[º], **Claudia Patricia Flórez Ramos** *[ID]*[º]

Plant Breeding, National Coffee Research Center (Cenicafé), Manizales, Caldas, Colombia

[º] These authors contributed equally to this work.

\* juancarlos.arias@cafedecolombia.com

**Data Availability Statement:** All relevant data are within the manuscript and its Supporting Information files.

## Abstract

In coffee (*Coffea arabica* L.), male sterility is a prerequisite for the exploitation of heterosis since it provides an efficient and reliable method for the production of hybrid seeds. Given its relevance, the objective of this study was to identify male-sterile genotypes within the Colombian Coffee Collection that can be used in genetic improvement. For this purpose, Ethiopian germplasm and progenies derived from hybrids between *C. arabica* x *C. canephora* were explored between 2017 and 2021. In the first stage, genotypes without visual presence of pollen were preselected in the field, followed by selection through staining and verification of male sterility and female fertility through directed crosses (directed, reciprocal and selfing). In this stage, 9,753 trees were explored, preselecting 2.4% due to visual absence of pollen. The staining of structures allowed us to confirm the lack or sporadic production of pollen in 23 individuals of Ethiopian origin. The results of the directed crosses led to the identification of 11 male-sterile and 12 partially male-sterile genotypes belonging to 15 accessions. In all cases, the individuals were characterized by the presence of anthers but with an absence or low content of pollen, which is why the male sterility is possibly of the sporogenic type. The female receptivity values were between 2.9% and 72.6%, being higher than 30% in five genotypes. These genotypes are a valuable tool for the genetic improvement of *C. arabica* with the potential to facilitate the use of heterosis and to allow a deeper understanding the development of male gametophytes in the species.

## Introduction

Coffee (*Coffea arabica*) is one of the most important raw materials worldwide, with an estimated annual market value of $200–250 billion, and is the main source of income for more than 60 million people [1]. Obtaining significant increases in agronomic traits of interest, especially yield, is one of the main objectives of genetic improvement programs, including for *C. arabica* given its economic importance. For this purpose, the use of the heterosis phenomenon is the main strategy employed in various species [2, 3]. However, the application of this strategy is conditioned by the finding of technical solutions at a moderate cost that allow it to be used commercially [4, 5]. This has been the major barrier for the use of heterosis in *C. arabica*,

**Funding:** This study was supported by The National Coffee Research Center (Cenicafé) (Crossref Funder ID 100019597). The funders had no role in study design, data collection and analysis, decision to publish, or preparation of the manuscript.

**Competing interests:** The authors declare that they have no competing interests.

given its floral morphology that favors autogamy, the manual seed collection-based methods used, such as the case for Ruirú 11 in Kenya [6], and the mass multiplication through somatic embryogenesis, such as the case for various hybrids in Central America [7], have failed to meet this premise.

Male sterility, a phenomenon that prevents the formation of functional male gametes, has been the tool par excellence used to take advantage of heterosis [8–13] and a prerequisite in the case of self-pollinated species [3, 12]. The phenomenon has multiple causes and may be the result of adverse growth conditions, diseases or mutations [8], the latter being one of major interest for genetic improvement. The development of the male organ and the formation and final release of pollen are made up of a series of perfectly synchronized stages [14] and failure in any of them can lead to male sterility [5, 15, 16]. The loss of male function has an important role in the evolution and adaptation of plants and is considered an intermediate step toward the separation of the sexes (dioecy), thus promoting cross-pollination [17]. Male sterility is infrequent but common in nature [18], and depending on the type of inheritance, it can be genetic (GMS) or cytoplasmic (CMS) [8, 9, 19].

In *Coffea* L., male sterility has been reported in diploid species [20, 21], interspecific hybrids [22], and in the most economically important species, *C. arabica* [23, 24]. In the latter, studies were carried out particularly on the germplasm from Ethiopia, the center of origin and diversity of the species. Despite this fact, information related to the knowledge of male sterility and mode of inheritance is scarce [23] and only recently used [25]. In this sense, the sterile male CIR-SM01, possibly related to the sources reported by Dufour *et al.* [24], was used to obtain the Starmaya® variety, comprising the first report of the use of male sterility in the genetic improvement of *C. arabica* [25].

In general, for the reported sources of male sterility, access and available information related to their use is restricted to the research centers where they have been identified, which has possibly limited their knowledge and use. Similarly, in Colombia, despite the availability of conserved germplasm in the Colombian Coffee Collection (CCC) and of information on genotype characterization and evaluation, as well as the reported potential of using heterosis and male sterility as a tool to facilitate genetic improvement, the search for male-sterile genotypes has not been carried out for *C. arabica*. Thus, the objective of this study was to identify sources of male sterility within the CCC for use in the genetic improvement program. The exploration carried out allowed us to recognize genotypes with this attribute, characterized by the presence of anthers without the presence of pollen, where female fertility is preserved.

## Materials and methods

### Study area

The genotypes used in this study belong to the CCC, which is established at the Naranjal Experimental Station of the National Coffee Research Center (Cenicafé), located at ecotope 206A in the municipality of Chinchiná (Caldas, Colombia) at geographical coordinates 4˚59'N, 7˚39'W. The necessary permissions for accessing the germplasm conserved at the CCC during the duration of the research were granted by the research center.

### Exploratory search of male-sterile genotypes

**Field preselection.** For the identification of male-sterile *C. arabica* genotypes, trees of Ethiopian origin and progenies between $F_2BC_2$ and $F_6BC_1$ originating from interspecific hybrids between *C. arabica* x *C. canephora*, were evaluated in the period from 2017 to 2021. The first group was introduced to the CCC between 1966 and 1989 as a result of the surveys conducted by the FAO [26] and ORSTOM (now IRD) [27] in Ethiopia. The second group was

obtained at Cenicafé with the objective of transferring resistance to coffee leaf rust (*Hemileia vastatrix* Berk & Br.) from *C. canephora* to *C. arabica*, through 1 or 2 backcrosses to the latter.

Coffee is a perennial species that has two flowering periods (December-April and May-August) in the study area, during which events of varying magnitude are observed. Thus, in the events that occurred between 2017 and 2018, at least 10 flowers in anthesis were collected from each individual, and each flower was rubbed on a black surface measuring 2.5 cm x 12 cm. In this way, potentially male-sterile plants were quickly identified due to the visual absence of pollen in all the observed flowers.

**Staining and detailed observation.**   For the preselected genotypes and onward, 10 to 15 flower anthers between stages 59 and 61 according to the BBCH scale for coffee [28], were collected for observation in greater detail under a microscope. At the time of their opening (12 hours later), the anthers were extracted, mounted on a slide and a drop of 1% acetic carmine was added, and slight pressure was applied to force their opening. The anthers were then removed, and the slide was observed under a microscope with a 10x objective. This procedure was repeated at least three times for two or three flowering periods. In the genotypes where no pollen or a reduced number of grains was observed, their phenotype was evaluated in search of symptoms indicative of sterility, such as the lack of fruits.

**Controlled pollination.**   In the genotypes without pollen formation or with a reduced amount of it, manual crossings and selfing were carried out to define two fundamental aspects: a) confirm the absence of pollen and b) determine the female receptivity of the genotypes, measured as the capacity to form fruits. In the principal flowering events that occurred during the flowering periods between 2019 and 2020, for each genotype, between 300 and 900 flower buds were selected for selfing (SF), directed crosses (DC—fertile x male-sterile) and reciprocal crosses (RC—male-sterile x fertile) using stigmas (previously emasculated flowers) and pollen, respectively, from the Castillo® General variety. Given the small size of the flowers, the manual pollen flow was achieved by of rubbing the anthers, still attached to the flower, on the stigmas of the receptor genotype, which was performed in all cases.

For each genotype, the pollination treatments were carried out between one and three occasions, depending on the availability of flowers in different flowering events. In all cases, the flowering branches were protected 48 hours before anthesis until 96 hours after the procedure was performed to avoid contamination with foreign pollen. Every two weeks until ripening, all the new flower buds were removed, and the final number of fruits formed in each of the treatments was recorded. Through directed crosses, a genotype was confirmed to be male-sterile when fruit formation occurred only when the genotype acted as the pollen receptor (RC).

To investigate the expression of the male sterility phenomenon in the species, one of the identified genotypes was selected to undergo a detailed examination of its anthers and female fertility, which were compared with those of the normal genotype from which it was derived or of genetically related genotypes.

**Morphological examination of anthers.**   In the selected genotypes, flower anthers were taken for observation of their structure in a scanning electron microscope (SEM) and examination by means of fluorescein diacetate (FDA) staining in a fluorescence microscope. For SEM observations, anthers 24 and 12 hours prior to anthesis and anthers of newly opened flowers were extracted and fixed in FAA until use. Subsequently, they were observed using a SEM. In the case of FDA staining, the procedure consisted of obtaining longitudinal and cross sections of mature anthers for both genotypes, followed by mounting on a slide containing 2 µg/ml FDA solution for 30 minutes, and later observed in an Nikon Eclipse 90i fluorescence microscope under ultraviolet light at 350–400 nm.

**Examination of female fertility.**   Female fertility was evaluated by observing the growth of the pollen tube under a MF according to the aniline blue fluorescence (ABF) method [29]

and by the formation of fruits in the field by directed manual crosses. In each plant, approximately 200 flower buds were emasculated and bagged at the time of anthesis, half of them were pollinated with pollen of the Caturra variety with 90% viability, while in the rest, directed and reciprocal crosses were performed between the male-sterile genotype and its parent. The emasculated and pollinated pistils were bagged immediately after the procedures. After 24 hours, 20 of the pistils pollinated with the Caturra variety were removed at random, fixed in FAA (10% formaldehyde:5% acetic acid:50% alcohol:35% distilled water) and stored at 4°C until use. The pistils where the directed and reciprocal crosses were performed were left on the plant. Likewise, at least 200 flowers without emasculation in both genotypes were protected for 96 hours to promote self-pollination, and 200 were marked and left exposed. In all cases, monitoring was carried out every 15 days for the elimination of new primordia, and the formation of fruits was recorded 120 days after pollination.

For the ABF method, the conserved pistils were 1. washed to remove any remaining FAA, 2. hydrolyzed with 8 N NaOH in a water bath for 20 minutes at 60°C, 3. washed again with water, and 4. submerged in aniline blue (1%, in 0.1 M $K_2HPO_4$) for 2 hours in the dark. Later, the cells were observed under a fluorescence microscope (Nikon Eclipse 90i) under ultraviolet light at 350–400 nm. A pistil was classified as receptive when at least one pollen tube reached the proximal growth position.

## Results

### Field preselection

In the Colombian Coffee Collection (CCC), approximately 10,000 trees were explored in search of this male sterility, 68.5% corresponding to 482 accessions from the center of origin and diversity of *C. arabica*–Ethiopia, while the remaining trees were derived from 79 progeny from the hybridization between *C. arabica* x *C. canephora*. This first step allowed reducing the genotypes to be evaluated in detail to only 2.43% of the population (Table 1) in which the presence of pollen was not observed with the naked eye. Of the preselected genotypes, 54.7% were of Ethiopian origin and 55.3% were derived from interspecific hybrids.

### Staining and detailed observation

The detailed microscope observation of the anthers of the genotypes selected in the field stage evidenced the reduced or no presence of pollen grains in 24 of them, 23 of which came from 15 Ethiopian accessions of the CCC and one from interspecific hybrids. The genotype 05.26/168*ms*463 was the only genotype in which the sporadic presence of viable pollen was observed. For the genotypes of Ethiopian origin, five accessions were characterized by having more than one individual with an apparent lack of pollen, highlighting CCC446, among which 20% of the total evaluated plants were selected.

The field observation of these genotypes showed a normal appearance in general; however, the genotype from interspecific hybridization had small flowers, irregular leaves and almost total absence of fruits, so it was discarded in the next stage, since the absence of pollen was possibly linked to total sterility or chromosomal aberrations.

**Table 1. Groups of explored genotypes and percentage selected at each stage.**

| Group | No. of evaluated plants | Participation (%) | Preselected in the field (%) | Preselected in the laboratory (%) | Male-sterile genotypes (%) |
|---|---|---|---|---|---|
| **Ethiopian origin** | 6,683 | 68.52 | 0.87 | 0.34 | 0.16 |
| **Interspecific hybrids** | 3,070 | 31.48 | 1.56 | 0.03 | 0.00 |
| **Total** | 9,753 | 100.00 | 2.43 | 0.24 | 0.11 |

## Controlled pollination

The results of crosses and selfing corroborated the absence or sporadic production of viable pollen in 52% of the preselected genotypes (Table 2). According to established criteria, 11 genotypes belonging to nine CCC accessions can be considered male-sterile given the nonformation of fruits in SF and DC, while fruit formation was observed in RC. In genotype 05.26/168*ms*463, the sporadic presence of viable pollen determined by staining was correlated with the data obtained at this stage, where 1.0% of flowers were effectively fertilized in DC, in contrast to SF, where fruit formation was not observed, possibly affected by the lack of manual pollen transfer. In the remaining 11 genotypes derived from nine accessions, it is important to note that although pollen was not observed, there was fruit formation either by SF or DC, which, as in 05.26/168*ms*463, were low, with values between 0.5% (SF—16.07/446*ms*248) and 5.6% (DC—05.26/480*ms*607). Similarly, while fruit formation was observed through both SF and DC for five of these genotypes, for 16.07/446ms158, 16.07/376ms382, 05.26/285ms425, and 05.26/393ms197, it occurred only through DC, and for 16.07/446ms149 and 16.08/254ms1486, it occurred only through SF. Given the production of pollen on some occasions in these 12 genotypes, they could be considered partially male-sterile.

Female fertility was highly variable among the genotypes, with an average of 31.9%, with a minimum value of 2.8% (16.08/386*ms*917) and a maximum value of 73.2% (16.07/446*ms*158).

**Table 2. Percentage of fruit formation in the individuals selected for having low or no pollen content in anthers.**

| CCC ID | Genotype | Percentage of Fruit Formation | | |
| --- | --- | --- | --- | --- |
| | | Selfing (SF) | Fertile x Male-sterile (DC) | Male-sterile x Fertile (RC) |
| 446 | 16.07/446*ms*565[*] | 0.0 | 0.0 | 46.3 |
| | 16.07/446*ms*020[*] | 0.0 | 0.0 | 21.0 |
| | 16.07/446*ms*149 | 1.0 | 0.0 | 61.7 |
| | 16.07/446*ms*158 | 0.0 | 0.5 | 73.2 |
| | 16.07/446*ms*248 | 0.5 | 0.5 | 54.1 |
| 376 | 16.07/376*ms*169 | 2.5 | 2.5 | 42.9 |
| | 16.07/376*ms*382 | 0.0 | 0.6 | 10.8 |
| 379 | 16.07/379*ms*762[*] | 0.0 | 0.0 | 72.6 |
| 285 | 05.26/285*ms*424[*] | 0.0 | 0.0 | 27.4 |
| | 05.26/285*ms*425 | 0.0 | 5.5 | 33.3 |
| 292 | 05.26/292*ms*734[*] | 0.0 | 0.0 | 32.4 |
| | 05.26/292*ms*740[*] | 0.0 | 0.0 | 43.6 |
| 343 | 05.26/343*ms*999 | 3.3 | 4.4 | 7.6 |
| 168 | 05.26/168*ms*463 | 0.0 | 1.0 | 26.6 |
| 194 | 05.26/194*ms*497[*] | 0.0 | 0.0 | 27.4 |
| 480 | 05.26/480*ms*607 | 5.0 | 5.9 | 38.1 |
| 195 | 05.26/195*ms*623[*] | 0.0 | 0.0 | 2.9 |
| 344 | 05.26/344*ms*2073[*] | 0.0 | 0.0 | 27.8 |
| 318 | 05.26/318*ms*2148[*] | 0.0 | 0.0 | 50.2 |
| 393 | 05.26/393*ms*197 | 0.0 | 1.9 | 23.9 |
| 386 | 16.08/386*ms*917 | 4.1 | 0.3 | 2.8 |
| | 16.08/386*ms*1333[*] | 0.0 | 0.0 | 16.0 |
| 254 | 16.08/254*ms*1486 | 2.8 | 0.0 | 25.9 |

[*]Genotypes considered male-sterile, since the percentage of fruit formation in the SF and DC controlled pollination treatments was equal to zero, while in RC, it was greater than zero.

This variation was also observed for plants with a common origin, such as CCC446, with values of 21.0% (16.07/446*ms*020) and 73.2% (16.07/446*ms*158), and CCC386, with 2.8% (16.08/386*ms*917) and 16.0% (16.08/386*ms*1333).

Similarly, among the accessions that presented more than one plant preselected due to the absence of pollen, CCC446, CCC285, and CCC386 had plants considered male-sterile and partially male-sterile, while the CCC292 and CCC376 genotypes were consistent in the absence and partial presence of pollen, respectively.

The results of directed pollination in the field may be affected by technical errors such as emasculation of mature buds, contamination with external pollen, or adverse climatic conditions, which affect the final decision to classify a genotype as totally or partially male-sterile or with high or low female receptivity. However, regarding the latter aspect, the replications in time led to similar results for genotypes with low receptivity, therefore, the possibility that not only pollen formation is compromised in these genotypes should be considered (S1 Table).

## Anther morphology

From the group of genotypes defined as male-sterile, 16.07/379*ms*762, along with a male-fertile plant from the accession from which it was obtained (CCC379/1032), were selected for detailed characterization. At first glance, the differentiation between the two genotypes is difficult, since the small size of the flower in the species makes it difficult to observe slight changes in its structure and the male-sterile genotype shows similar floral characteristics to those of the normal genotype (Fig 1A and 1B), which was common for all male-sterile genotypes identified in this study. In greater detail, in the male-fertile genotype, the anthers were yellowish with a cottony texture (Fig 1C), and some pollen grains were observed (Fig 1E). In the male-sterile genotype, the anthers were whitish with a smooth texture (Fig 1D), and no pollen content was observed (Fig 1F).

The SEM images for the three anther development stages analyzed (24 and 12 hours prior to anthesis and in anthesis) show considerable differences between the genotypes (Fig 2). In principle, in anthers from the normal genotype, at 24 hours prior to anthesis the turgor of the organ is evident, and it is possible to observe the beginning of the opening of the sacs and the presence of pollen grains (Fig 2A). In contrast, anthers in this same state from the male-sterile genotype show less turgor and no clear separation of the pollen sacs that comprise it (Fig 2B).

The anther dehiscence process is more noticeable in flowers 12 hours prior to anthesis, a stage better known as "candle", characterized by the white hue of the petals that are still closed. During this stage in the normal genotype, the anthers are completely open and contain a considerable number of exposed pollen grains, and the anther's own tissue are starting to undergo senescence, which is noticeable by the loss of turgor (Fig 2C) and more accentuated when the flower is fully open (Fig 2E). In contrast, in 16.07/379*ms*762, loss of turgor of the anther is observed prior to the flower opening event (Fig 3D), which continues until this last stage (Fig 2F), with no notable differences between the stages.

The absence of pollen is clear in the anthers of 16.07/379*ms*762 when they are observed under a fluorescence microscope (Fig 3). In the normal genotype (CCC379/1032), staining with FDA indicated the presence of pollen grains uniformly distributed in the tissue (Fig 3A and 3C), while in anthers of 16.07/379*ms*762, there were no grains (Fig 3B and 3D).

## Female fertility

The results obtained indicate that the absence of pollen in 16.07/379*ms*762 is a consequence of male sterility. The observations made by the ABF method (Fig 4) indicate that the female fertility of 16.07/379*ms*762 is preserved, since no differences were identified in the growth of the

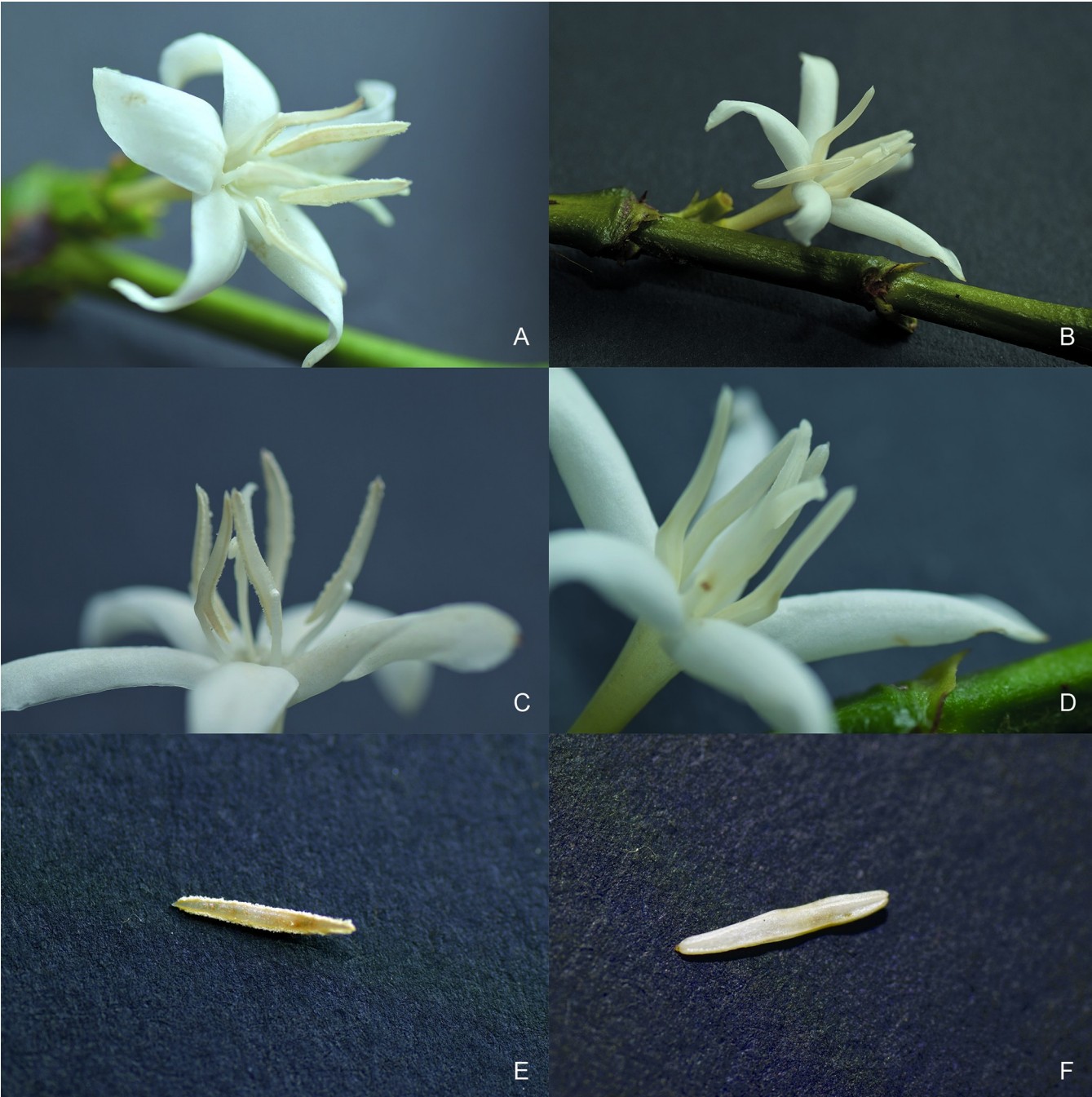

**Fig 1. Morphological comparison of the floral structures of CCC379/1032 (male-fertile) and 16.07/379*ms*762 (male-sterile).** Flower appearance of (A) CCC379/1032 and (B) 16.07/379*m*762, with no differences identified with the naked eye. (C and E) Anthers of CCC379/1032 with rough texture and yellowish color, where open pollen sacs and the presence of pollen are observed. (D and F) Anthers of 16.07/379*ms*762 with smooth texture and whitish color and without the presence of pollen.

pollen tube between pistils of CCC379/1032 (normal) and 16.07/379*ms*762 (male-sterile). In both genotypes, a considerable number of pollen tubes were observed near the stigma (Fig 4A and 4B), of which, in all the observed pistils, at least one reaches the proximal growth position (Fig 4C and 4D).

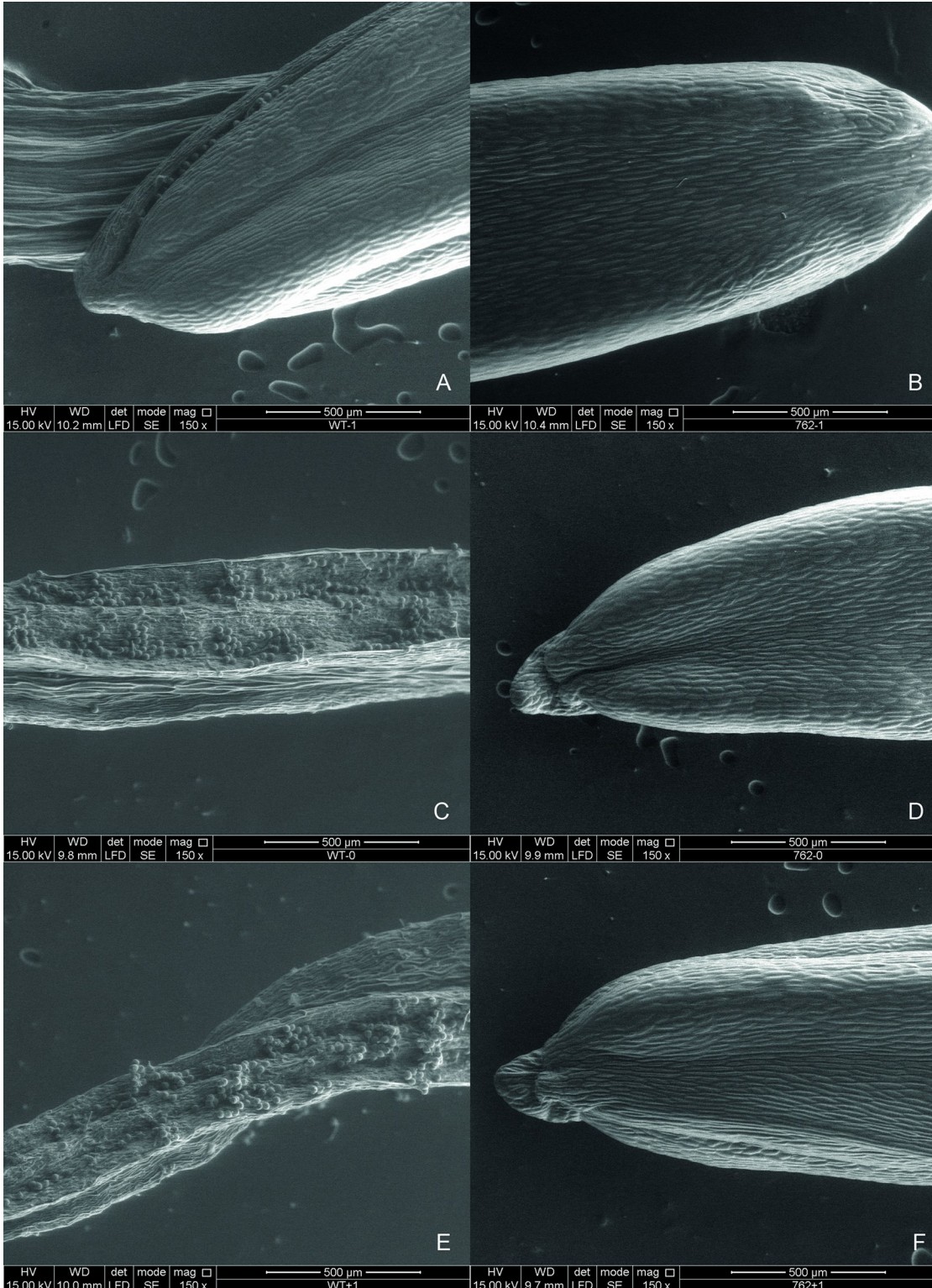

**Fig 2. Morphological comparison of anthers of CCC379/1032 (male-fertile) and 16.07/379*ms*762 (male-sterile).** (A) Anther 24 hours prior to anthesis in CCC379/1032, showing the beginning of pollen sac opening. (B) Anther 24 hours prior to anthesis at 16.07/379*ms*762 with no opening of pollen sacs. (C) Anther 12 hours prior to anthesis in CCC379/1032, fully open and releasing pollen. (D) Anther 12 hours prior to anthesis in 16.07/379*ms*762, with slight loss of turgor and no opening. (E) Anther in anthesis in CCC379/1032, with pollen release. (F) Anther in anthesis in 16.07/379*ms*762, with loss of turgor and no pollen release.

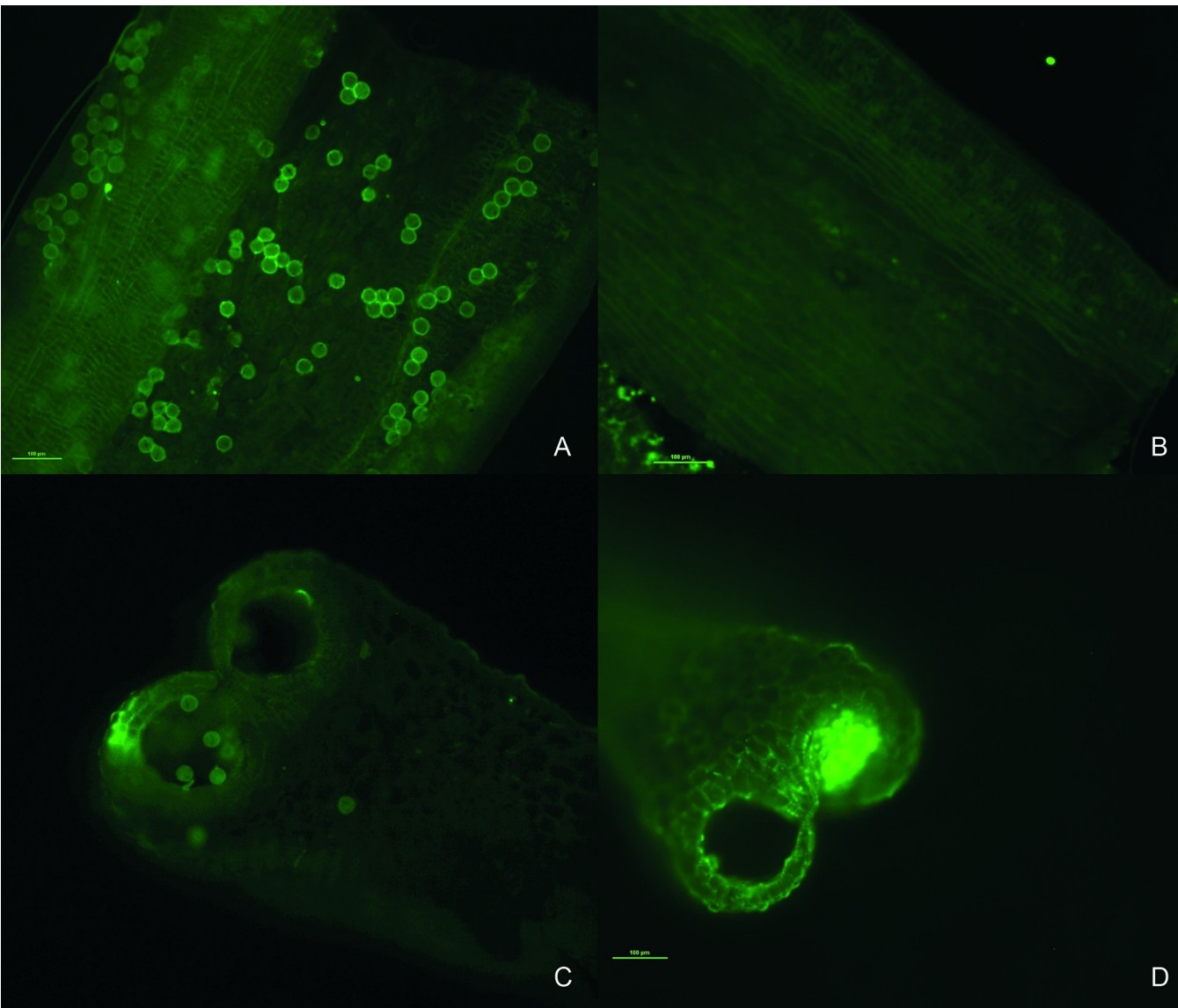

**Fig 3. Morphological analysis of pollen of CCC379/1032 (male-fertile) and 16.07/379*ms*762 (male-sterile).** (A) Longitudinal section of an anther of CCC379/1032 showing viable pollen grains within the lobes. (B) Longitudinal section of an anther of 16.07/379*ms*762 without the presence of pollen grains in the lobes. (C) Cross-section of an anther of CCC379/1032 showing viable pollen grains in the lobes. (D) Cross-section of an anther of 16.07/379*ms*762 without the presence of pollen grains in the lobes. Bars = 100 μm.

Under field conditions, the flowers in the male-sterile genotype (16.07/379*ms*762) were isolated to force self-pollination but no fruits were formed (Fig 5A). In contrast, in the male-fertile genotype (CCC379/1032), 68.2% of the isolated flowers formed fruits (Fig 5B). Female fertility in 16.07/379*ms*762 was corroborated, since fruits were obtained in 63.5% and 40.5% of flowers manually pollinated and freely exposed, respectively (Fig 5C and 5E). In the case of CCC379/1032, when anthers of the male-sterile genotype were used to fertilize its flowers, no fruits were formed (Fig 5D), while in free pollination, the percentage of flowers with fruit formation was higher than 75% (Fig 5F). This finding reinforces the absence of pollen in 16.07/379*ms*762 and the importance of the presence and abundance of pollinators to carry out pollen transfer. These results are consistent with the initial observations made in the identification of this genotype.

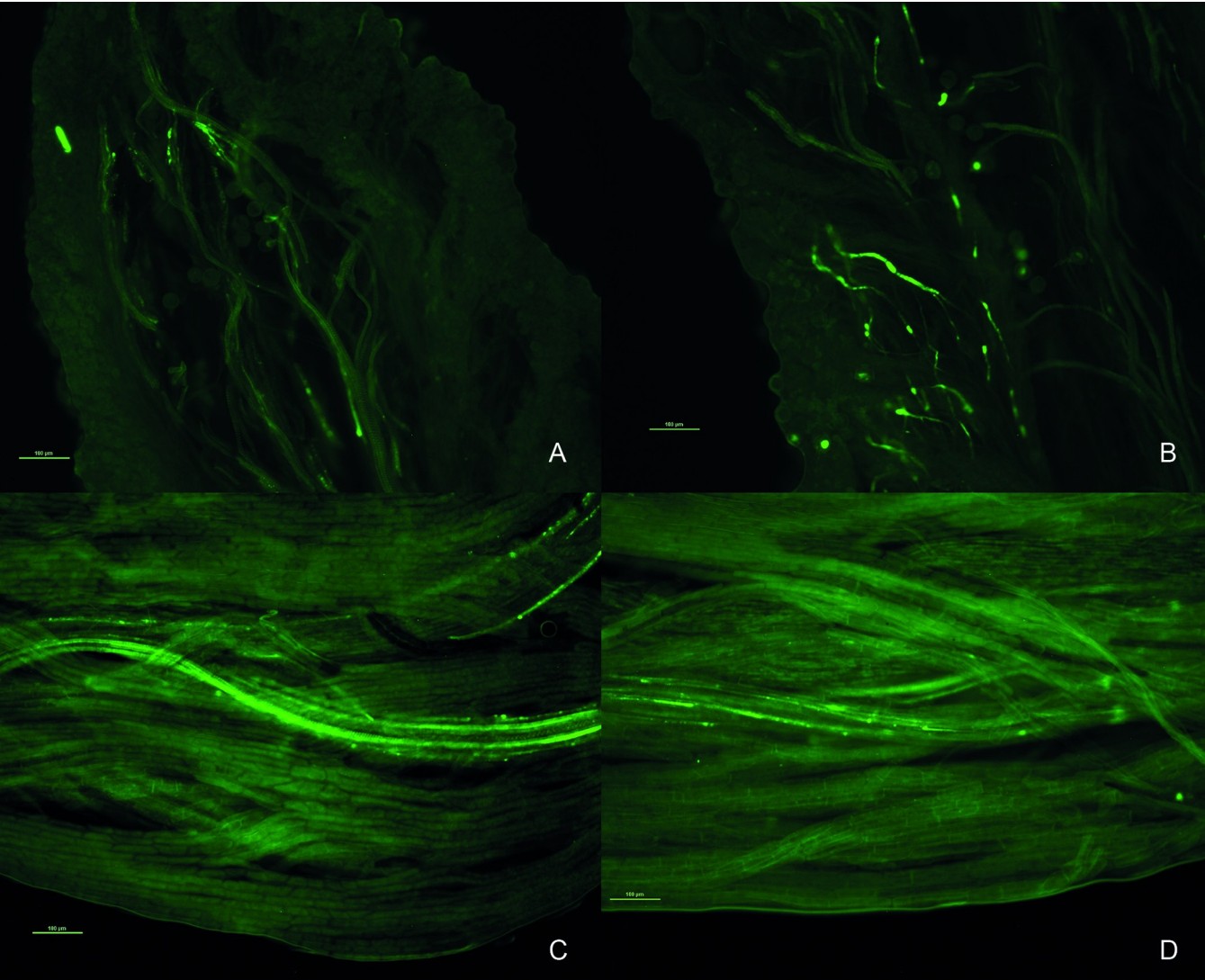

**Fig 4. Pollen–pistil interaction in CCC379/1032 (male-fertile) and 16.07/379*ms*762 (male-sterile).** (A) Pollen grains and generation of pollen tubes in stigma of CCC379/1032. (B) Pollen grains and normal generation of pollen tubes in stigma of 16.07/379*ms*762. (C) Presence of pollen tubes in the proximal growth position, close to the ovary, in CCC379/1032. (D) Presence of pollen tubes in the proximal growth position, close to the ovary, at 16.07/379*ms*762. Bars = 100 μm.

Having genetic diversity is the main tool for genetic improvement programs, since it offers the possibility of finding traits of interest to solve current or potential problems in the species of interest. In this case, the 11 male-sterile genotypes and the 12 partially male-sterile genotypes identified provide the possibility of evaluating and selecting those with the highest agronomic value and using them to facilitate the exploitation of heterosis in coffee mediated by male sterility.

## Discussion

For *C. arabica*, the use of F1 hybrids has the potential to achieve significant increases in agronomic traits of interest, thus being a promising alternative to traditional cultivars [25]. However, its success, as in any species, depends on having a technical solution at a reasonable price

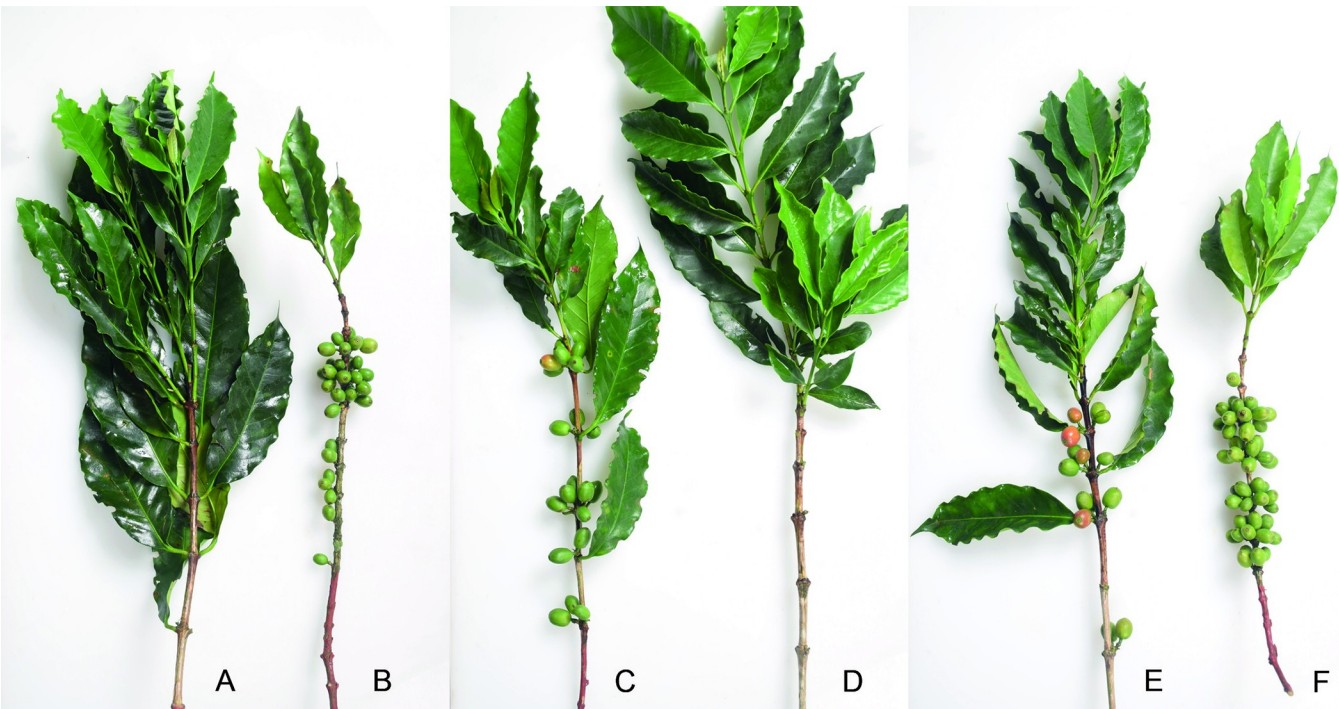

**Fig 5. Formation of fruits by directed crosses and free pollination in CCC379/1032 (male-fertile) and 16.07/379*ms*762 (male-sterile).** (A) No fruit formation in 16.07/379*ms*762 with isolated flowers. (B) Fruit formation by selfing in CCC379/1032 with isolated flowers. (C) Fruit formation in 16.07/379*ms*762 with manual pollen transfer from the male-fertile genotype. (D) No fruit formation in flowers pollinated with "pollen" from the male-sterile genotype. (E) Fruit formation by free pollination in the male-sterile genotype. (F) Fruit formation by free pollination in the male-fertile genotype.

for commercialization [4, 5]. It is of little use to find an excellent hybrid combination if there is no practical and economical way to exploit it [5]. For *C. arabica*, this has been the greatest limitation, given the self-pollinating nature of the species, and the methodologies currently used for the distribution of F1 hybrids have a significantly higher technical and economic costs compared to seed-propagated varieties [30, 31]. In this sense, despite the great achievements obtained in multiplication via somatic embryogenesis [7], its high cost [31], in addition to the necessary specialized workforce and infrastructure, continues to be one of the greatest obstacles to its use.

Male sterility, a phenomenon that prevents the formation of functional male gametes, has been especially useful to overcome the limitations in the commercial production of hybrids [3, 32]. In this study, we report for the first time for Colombia the identification of 11 male-sterile genotypes that could be used in genetic improvement. This is not a new finding for the genus *Coffea*, since genotypes with this trait have long been reported [20–24] but no deep knowledge about it. The male sterility, despite being a rare phenomenon [8, 18], it has been repeatedly observed in the wild germplasm of *C. arabica* [23, 24], antecedents drove the initiative to explore the CCC in the search for sources of male sterility, since some of the reports were of germplasm common among research centers. Of the six male-sterile genotypes reported by Mazzafera *et al.* [23] in the germplasm bank of the Agronomic Institute of Campinas (IAC), Brazil, four are related to Ethiopian germplasm, three of them were found among plants of the C.2154 accession from the FAO survey. Likewise, Dufour *et al.* [24], within the Coffee Collection of CATIE, Costa Rica, identified five male-sterile plants in four accessions from the same survey: T4601 (E.343), T4621 (E.123b), T4759 (E.238 and E.328?), and T4905 (E.536). Although the genotypes identified in this study have the same origin (FAO survey),

only one accession (E.238, and possibly E.328) coincides with the findings of Dufour *et al.* [24], and they may differ from the reports of Mazzafera *et al.* [23]. Given these findings, two aspects of male sterility in *C. arabica* are important to note: the high frequency of male function loss in wild germplasm is confirmed, raising interest in studying its role in the process of evolution and adaptation. Second, there is a potential effect of the germplasm collection objective on this characteristic, as the characteristic is almost exclusively found in the accessions collected by the FAO. In this sense, while the FAO survey aimed to collect the greatest possible genetic diversity for *C. arabica* [26], the study carried out by Guillaumet & Halle for ORSTOM concentrated on genotypes with agronomic value that could be directly used [27], which notably reduced the possibility of finding male-sterile plants.

The exploration carried out in this study also allowed the identification of partial male sterility for the first time in coffee, where for 11 genotypes, the sporadic presence of pollen was inferred from the results of directed crosses. However, given the impossibility of observing pollen in most of these cases, in the four instances where fruit formation occurred only through DC, it is important to take certain considerations into account. In the DCs, flower buds of fertile plants of the Castillo® variety were emasculated to avoid their self-pollination. However, in this state, the recognition of flower buds in which the pollen sacs have not yet opened is difficult, which may lead to technical errors [33], and up to 6.3% fruit formation in previously emasculated flowers was observed [34]. The SEM observations explain this because it was possible to observe the beginning of pollen sac opening in anthers before the occurrence of anthesis. Therefore, it is possible that in the DCs performed with 16.07/446*ms*158, 16.07/376*ms*382, 05.26/285*ms*425 and 05.26/393*ms*197, where there was fruit formation by DC but not by selfing (SF), the occurrence of self-fertilization before emasculation in the genotype used as the female parent is not ruled out. Considering this fact, along with the instances where zero values were obtained in DC for these genotypes, it is possible that a condition of total androsterility is present, necessitating further investigation into their behavior and expanding the number of available sources. The opposite case are the genotypes in which there was formation of fruits by SF since the time elapsed between the isolation of the flowers (48 hours before anthesis) and collection (96 hours after anthesis), added to the uniformization of the ages of the flowers, prevents contamination by foreign pollen; hence, sporadic pollen production is possible. Although Duffour *et al.* [24] considered as contamination the fruits formed by SF in two of the genotypes they deemed male-sterile, in our case, given the precautions taken and the importance of the total absence of pollen for the purity of the seeds produced, this concept was not adopted.

According to Kaul [35], completely male-sterile genotypes, such as those identified in this study, hold the highest value for plant breeding. Nevertheless, various studies have demonstrated the utility of partial male-sterility, provided that the conditioning factors have been identified [36]. In this regard, its application has been promoted due to simplification of the multiplication process of the male-sterile source and hybridization schemes [3–5, 9, 37]. Hence, the considerable numbers of totally and partially male-sterility sources currently available for coffee genetic improvement would allow for the exploration and selection of those with the greatest value. However, given the perennial nature of the species, the use of the second type may have certain limitations unless a marker is developed to enable early selection of seeds or genotypes resulting from self-pollination.

As in wild germplasm, there were previous reports of male-sterility in hybrids derived from interspecific hybridization [22], because the interactions between the genomes, cytoplasm, and nuclei of distant species usually cause aberrations that can affect the reproductive organs [35, 37]. Moreover, in addition to the existing differences in ploidy between *C. arabica* (4x = 44) and other species in the genus (2x = 22), which could favor such events, this could explain the

findings of Vishveswara [22]. However, among the explored germplasm of this origin, no genotypes were found to be male-sterile, possibly due to the intrinsic characteristics of the population used. This population was created using the pathway of triploid hybrids, where one or two backcrosses to *C. arabica* are sufficient to recover ploidy and fertility, with severe counter-selection against introgression from the diploid species [38], and possibly any negative effects of hybridization. Furthermore, the genotypes evaluated in this study belong to advanced generations ($F_2BC_2$ and $F_6BC_1$) subjected to continuous evaluation and selection processes, which possibly eliminated individuals with some biological disadvantage such as the absence of pollen. These selection processes would also explain the absence of male sterility in cultivated germplasm from other origins [23, 24], including those of Ethiopian origin with some degree of selection (ORSTOM survey) explored in this study. It is possible that in early generations of interspecies hybrid populations, male-sterile plants can be found; however, the differences in ploidy mentioned affect other aspects related to fertility and agronomic value, which requires a long period of evaluation and selection, limiting their use.

Although the discovery of genetically emasculated genotypes is key to facilitating the commercialization of F1 hybrids, there are limitations to their use when female fertility is also affected [31]. According to Graybosch & Palmer [15], the mutations that cause the absence of pollen or male sterility in an individual can also have an extensive, slight or absent effect on female fertility. In general terms, any negative effect on female fertility will have a direct impact on the amount of seeds produced and consequently their cost, so a true male sterile genotype should not present any limitation, and when is affected, it may be the consequence of meiotic mutations or complete sterility [35]. Despite the importance of this aspect, it is little specified in the different previous reports, including those for coffee. In this study, the female fertility of genotypes defined as androsterile or partially male-sterile was determined by the percentage of fruit formation in controlled crosses, showing significant variation among them. However, it is important to consider that in coffee, fruit formation by manual crossing presents high variation, and can be at least 13% [33], significantly lower than that found by Herrera & Gonzales [39], who obtained frequencies higher than 29%, a technically accepted value for this procedure when varieties of different origins are used (personal observation). In the case of the male-sterile genotypes identified in CATIE, the female fertility values were between 26% and 60% [24], a range observed in eight of the 11 male-sterile genotypes reported here. Given these considerations, these genotypes could be considered true male-sterile, presenting percentages of fruit formation by manual crossing slightly lower (05.26/194*ms*497) or higher than 29%. On the other hand, for 16.08/386*ms*1333, 05.26/195*ms*623 and 16.07/446*ms*020, negative effects as well as the absence of pollen being related to the factors described by Kaul [35] are not ruled out, given the low fruit formation observed in directed manual crossing.

Based on the detailed observations of 16.07/379*ms*762, it can be inferred that the type of male sterility is sporogenic (sterile pollen), a possible condition for the other genotypes identified, given the presence of normal anthers but with the absence of pollen [35, 40]. For *Coffea*, this is consistent with Mazzafera *et al.* [23] but differs from the observation by Georget *et al.* [25], who identified the functional type of male sterility. Regarding the mode of inheritance, the state of knowledge of the male sterility phenomenon is too incipient to reach a conclusion for the genotypes identified here; however, the observations made indicate that pollen production is completely restored in F1, suggesting that it is likely to be genetic (GMS) and recessive. Few studies have been carried out for *Coffea* on this subject, but they agree that male sterility is conditioned by the action of recessive genes [23–25], which does not depart from the general observations of the phenomenon in nature [19].

Despite its simplicity, the use of GMS has been limited by the difficulty of its mass multiplication or recognition in segregating populations. However, for *C. arabica*, the limitations for

the application of GMS can be easily overcome given the mass multiplication techniques available [7, 31]. Therefore, the genotypes identified here could be used in the commercial production of hybrids, as is the case for CIR-SM01 [25]. Similarly, given the perennial nature of the species, its multiplication by segregating F2 populations with subsequent elimination of fertile plants can also be considered, as is done in other species [13, 36].

However, beyond these barriers, it should be noted that in selfing species such as *C. arabica*, fruit formation in male-sterile plants is dependent on vectors with the ability to transfer pollen [36], unlike in male-fertile genotypes, where selfing occurs without their presence, or in outcrossing species, where pollen flow happens naturally. This was reflected in the percentage of fruits formed by natural pollination in 16.07/379$ms$762, which despite not presenting limitations in its female fertility, only 40% of the flowers were fertilized, while in the normal genotype, this figure exceeded 70%. In this sense, the experience of using male-sterile CIR-SM01, the progenitor of Starmaya [25], may reflect this same difficulty. In this case, a production between 62 and 94 grams of seed tree per year by natural pollination was reported, equivalent to approximately 125–190 fruits. This figure could be considered low if taking into account that, under similar environmental conditions, it represents between 5–10% of the potential in a traditional variety. However, the predominant factors that could have influenced these results in the study are unknown, and may be factors external to the genotype, such as a deficient donor plant scheme, deficiency in pollen vectors, adverse weather conditions, diseases, or factors specific to the genotype, such as a low productive potential (agronomic value) or the association between the absence of pollen and low female fertility [15, 35]. Despite this aspect, Georget *et al.* [25] highlight the feasibility of using the phenomenon for the commercial-scale production of F1 hybrids in *C. arabica*, emphasizing that the limited availability of sources is perhaps the greatest constraint, as it reduces the number of potential hybrids that can be created.

Given the existing need for pollen flow in self-pollinating species and the availability of sources of male sterility, a parameter that could be characterized is related to the traits that may limit or favor it in *C. arabica*. According to Neustupa & Woodard [41], the genes causing male sterility may have a strong influence on changes in the floral morphology of the species, especially traits that may play an important role in increasing its preference by pollinators [42, 43]. For the genotypes identified here, at first glance, no significant morphological changes between male-fertile and male-sterile genotypes were observed. In this sense, characterizations related to the floral morphology of *C. arabica* are scarce; however, a high environmental influence is recognized [44]. Given their importance for the use of male sterility in the species, detailed studies would be valuable to define strategies and the selection of the best sources to be used by genetic improvement programs.

The wild germplasm of *C. arabica* has generally been rarely used in genetic improvement programs; however, there is consensus on its potential [25, 30], so explorations of agronomically relevant traits and current utilization have gained interest. In this sense, its use in obtaining F1 hybrids has demonstrated particular utility, with observed values of heterosis ranging from 34% to 58%, showing potential not only to significantly increase yields but also to enhance sensory quality [30]. The exploration carried out allowed the identification of a significant number of sources of male sterility and partial male sterility for *C. arabica* among the Ethiopian germplasm conserved in the CCC. Given previous studies carried out since the introduction of the genotypes to the CCC, there is information on their agronomic traits, genetic diversity and population structure, which increases their value for their use (S2 Table). Similarly, although it is clear that cytological and genetic studies are important for understanding the phenomenon, they are not decisive for exploiting male sterility, thus genotype evaluation activities should be implemented.

## Conclusions

The male sterility phenomenon has undoubtedly marked a milestone for the genetic improvement of various species by facilitating the large-scale production of hybrid seeds, especially in self-pollinated plants, where conventional methods are not technically or economically viable. In coffee, despite its recorded existence, male sterility has not been widely used, but without a doubt, it has the potential to overcome the existing limitations for the commercialization of hybrids of the species.

The genotypes identified in the CCC have important applications in genetic improvement. First, they can be easily used in crosses with genetically diverse elite genotypes to evaluate their general and specific combinatory ability for selection. Second, they allow the evaluation of seed production potential under certain sowing schemes, a parameter that directly impacts the cost of commercialization. Third, through current biotechnological tools, it is possible to associate molecular markers, identify the gene or genes responsible for male sterility, and genetically modify genotypes of high agronomic value. Lastly, and perhaps the greatest benefit, is that the identified genotypes can lead in the long term to the use of F1 hybrids in coffee growing.

## Supporting information

**S1 Table. Results from controlled crosses.** https://doi.org/10.38141/10799/dataset03. (XLSX)

**S2 Table. Agronomic traits and sensory quality of the accessions for which plants with complete or partial male sterility were identified.** https://doi.org/10.38141/10799/dataset04. (XLSX)

## Acknowledgments

We thank C. Vera and G. Rodríguez for their help with the field activities and J. Quintero and H. Cortina for curating the Collection over time.

## Author Contributions

**Conceptualization:** Juan Carlos Arias Suárez, Claudia Patricia Flórez Ramos.

**Investigation:** Juan Carlos Arias Suárez.

**Methodology:** Juan Carlos Arias Suárez, Claudia Patricia Flórez Ramos.

**Supervision:** Juan Carlos Arias Suárez.

**Visualization:** Juan Carlos Arias Suárez, Claudia Patricia Flórez Ramos.

**Writing – original draft:** Juan Carlos Arias Suárez.

**Writing – review & editing:** Juan Carlos Arias Suárez, Claudia Patricia Flórez Ramos.

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
