## [Decision Letter · Decision Letter 0]

22 Jun 2023

PONE-D-23-12131Identification of sources of male sterility in the Colombian Coffee Collection for the genetic improvement of Coffea arabica L.PLOS ONE

Dear Dr. Arias,

Thank you for submitting your manuscript to PLOS ONE. After careful consideration, we feel that it has merit but does not fully meet PLOS ONE’s publication criteria as it currently stands. Therefore, we invite you to submit a revised version of the manuscript that addresses the points raised during the review process.

We look forward to receiving your revised manuscript.

Kind regards,

Abhishek Bohra

Academic Editor

PLOS ONE

Journal Requirements:

“This study was supported by The National Coffee Research Center (Cenicafé) (Crossref Funder ID 100019597)”

5. Please remove your figures from within your manuscript file, leaving only the individual TIFF/EPS image files, uploaded separately. These will be automatically included in the reviewers’ PDF.

Additional Editor Comments:

Experts have provided their feedback on the manuscript. I encourage authors to submit a revision based on the suggestions offered by the two Reviewers. Please find below the Reviewer's comments:

Reviewers' comments:

Reviewer's Responses to Questions

**Comments to the Author**

1. Is the manuscript technically sound, and do the data support the conclusions?

Reviewer #1: Yes

Reviewer #2: Partly

2. Has the statistical analysis been performed appropriately and rigorously? 

Reviewer #1: N/A

Reviewer #2: N/A

3. Have the authors made all data underlying the findings in their manuscript fully available?

Reviewer #1: Yes

Reviewer #2: No

4. Is the manuscript presented in an intelligible fashion and written in standard English?

Reviewer #1: Yes

Reviewer #2: Yes

5. Review Comments to the Author

Reviewer #1: The article needs to be improved with respect to clarity of expression. The article can be reduced further by removing the generalized statements which are not related to the aspects studied in this study. Particularly in the discussion part for example results of studies on the inheritance of male sterility have been given which is not being studied in this study. Thus, Discussion part should be reduced by removing the detailed information on inheritance aspect. The focus of the Results and Discussion should be only on the analysis of the results obtained for identification of new male sterile genotypes and their characteristic features and how they can be utilized further.

Suggested corrections may be incorporated, comments may be responded and article may be improved and resubmitted.

Reviewer #2: Reviewer’s comments PONE- D 23-12131

General observations:

(i) This paper reports the finding of a male sterility system, arising naturally

and through wide hybridizations. Besides, some partial male sterile plants

were also found. Various morphological details related to flower and anther

types are described well. The authors also suggest its potential use in the

genetic enhancement of coffee yields. Since this is the first such discovery,

it recommend its publication. (ii) At more than one places there are

repetitions of subject and text lines. Authors need to consider editing the

text.

Scientific observations.

The authors should make clear in the abstract, objective, introduction that

this paper deals with a genetic male sterility (GMS). It is better that the text

portions be slightly modified accordingly.

At present the discussion portion appears quite weak. It should be

strengthened by incorporating information on the issue of its potential

utilization in crop improvement, particularly in view of its autogamous

breeding nature. Although the authors have provided information on this

subject using some passing references, but, I feel, the paper will be

complete in itself if brief information on the large-scale seed production the

male steriles / hybrids and their limitations in coffee are included. Besides

information on the reported heterosis for yield will add to the overall quality

of the paper. And finally, is there any adverse effect on the quality of

beans?

Recommendation

This manuscript may be published after the revision.

6. PLOS authors have the option to publish the peer review history of their article (what does this mean?). If published, this will include your full peer review and any attached files.

Reviewer #1: No

Reviewer #2: No

---

## [Author Response · Author response to Decision Letter 0]

4 Aug 2023

Response to editor comments:

Journal Requirements:

Response: The manuscript complies with the requirements of the journal.

Response: A brief statement regarding the access requirements provided by the Research Center has been included in the manuscript.

“This study was supported by The National Coffee Research Center (Cenicafé) (Crossref Funder ID 100019597)”

Response: The role of the funding agency is included in the cover letter

Response: The DOIs will be included

5. Please remove your figures from within your manuscript file, leaving only the individual TIFF/EPS image files, uploaded separately. These will be automatically included in the reviewers’ PDF.

Response: Figures removed within the manuscript.

Response: Captions for supporting information were included according to the journal's format. 

Additional Editor Comments:

Experts have provided their feedback on the manuscript. I encourage authors to submit a revision based on the suggestions offered by the two Reviewers. Please find below the 

Reviewer's comments:

Reviewer comments #1: 

Field preselection

Reference Line No 87-89

How the F2BC2 and F6BC1 were derived should be made more clear. Which species was used as male parent for generating Back cross generations? 

Response: A clarification is made in the manuscript.

Reference Line No 94-97 

For carrying out staining and detailed observation

Of the preselected genotypes how many floral buds/ flowers per genotype were sampled 

for observation in greater detail under a microscope.

Response: A clarification is made in the manuscript.

In line no 100 &101 

For observation in greater detail under a microscope procedure was repeated at least three times for two or three flowering periods. Here flowering period should be defined.

Response: the term "flowering period" is mentioned three times, and its definition is provided in the first occurrence.

Controlled pollination

Line No 108:

How directed crosses (DC - fertile x male-sterile) were generated using male sterile parent as male parent?

Response: A clarification is made in the manuscript.

Line No 116:

In the DC (DC - fertile x male-sterile) male parent used is “male sterile” so there is no question of fruit formation ! Please clearify.

Response: When well-defined male-sterile plants are used, the outcome in the DC (DC - fertile x male-sterile) may seem obvious, as well as in the case of self-fertilization. However, the male-sterile condition in the preselected genotypes had not been confirmed, so it was possible for fruits to form, as observed in ten out of the 23 genotypes used.

A clarification is made in the manuscript.

Examination of female fertility

Line No. 135:

How male sterile genotype was used as male parent iin case of reciprocal crosses.

Response: Given the defined male-sterile condition, in the reciprocal crosses, it acted as the pollen receptor. Despite the natural absence of pollen, emasculation was performed to equalize the treatment with the genotype used as a control, so that any negative effect caused by this practice would affect both genotypes equally.

Staining and detailed observation

Line No 170-172:

 Absence of pollen 

In the study, genotype from interspecific hybridization had small flowers, irregular leaves, absence of pollen and almost total absence of fruits should not have been discarded. These should have been pollinated with the pollen from male fertile genotypes in order to identify the male sterile or maintainer genotype.

Response: The characteristics of the mentioned plant indicated that beyond male-sterile, the absence of pollen was caused by total sterility of the plant, possibly due to chromosomal aberrations given its hybrid origin. In addition, this genotype, like the other plants used in this study, was surrounded by fertile plants, so if it were male-sterile, the absence of fruits would not have been highly compromised. No modification is made to the manuscript.

Line No 322 to 332:

More clarity of expression is required.

Response: The sentence is reframed in the revised manuscript. A bibliographic reference is updated in this section.

Modified: Centro Nacional de Investigaciones de Café Informe Anual Cenicafé 2021 Colombia: El Centro; 2021, by: Gómez JH, Benavides P, Maldonado JD, Jaramillo J, Acevedo FE, Gil ZN. Flower-Visiting insects ensure coffee yield and quality. Agriculture. 2023;13: 1392. doi:10.3390/agriculture13071392

Line No 341 to 343 :

Meaning of the sentence is not clear.

Response: A sentence is removed, and a paragraph is relocated to enhance the clarity of the discussion.

Line No 363 to 365: 

Meaning of the sentence is not clear.

Whether following was to be stated !

Crosses between C. arabica (4x = 44) and other diploid species of the genus (2x = 22) leads to formation of triploids (3x = 33) that occasionally produce fruits, but after one or two back crosses with C. arabica, ploidy and fertility are recovered [37].

Response: It is adjusted in the manuscript to improve the clarity of the sentence.

Recommendation:

The article needs to be improved with respect to clarity of expression. The article can be reduced further by removing the generalized statements which are not related to the aspects studied in this study. Particularly in the discussion part for example results of studies on the inheritance of male sterility have been given which is not being studied in this study. Thus, Discussion part should be reduced by removing the detailed information on inheritance aspect. The focus of the Results and Discussion should be only on the analysis of the results obtained for identification of new male sterile genotypes and their characteristic features and how they can be utilized further.

 Suggested corrections may be incorporated, comments may be responded and article may be improved and resubmitted.

Response 

We sincerely appreciate the careful attention given to the review of our manuscript; the valuable suggestions provided have undeniably contributed to enhancing its clarity, particularly for readers who may not be familiar with the species. As a result of the feedback received, irrelevant sections of the discussion were omitted, and the manuscript was further strengthened by emphasizing the significance of the obtained results. Additionally, we have now incorporated a comprehensive analysis of the advantages and limitations related to the utilization of male-sterile and partially male-sterile genotypes in our study.

Revisor #2: 

General observations:

(i) This paper reports the finding of a male sterility system, arising naturally and through wide hybridizations. Besides, some partial male sterile plants were also found. Various morphological details related to flower and anther types are described well. The authors also suggest its potential use in the genetic enhancement of coffee yields. Since this is the first such discovery, it recommend its publication. (ii) At more than one places there are repetitions of subject and text lines. Authors need to consider editing the text.

Scientific observations.

The authors should make clear in the abstract, objective, introduction that this paper deals with a genetic male sterility (GMS). It is better that the text portions be slightly modified accordingly.

At present the discussion portion appears quite weak. It should be strengthened by incorporating information on the issue of its potential utilization in crop improvement, particularly in view of its autogamous breeding nature. Although the authors have provided information on this subject using some passing references, but, I feel, the paper will be complete in itself if brief information on the large-scale seed production the male steriles / hybrids and their limitations in coffee are included. Besides information on the reported heterosis for yield will add to the overall quality

of the paper. And finally, is there any adverse effect on the quality of beans?

Recommendation

This manuscript may be published after the revision. 

Response 

Thank you for the time taken to review our manuscript and for the important observations provided. Below, we are presenting our response to the various suggestions: 

For the sections that we considered repetitive, we unified them into a single paragraph within the discussion. Similarly, in some cases, sentences were relocated to improve clarity and enhance coherence between sections.

While it is suggested to specify in different sections of the manuscript that we are dealing with genetic male-sterility (GMS), the suggestion is only included in the part of the discussion where we mention, based on our observations, that fertility is reiterated in the F1 generation. In general, further investigation into this aspect is underway, as it is the first time that a considerable number of sources have been available to do so.

The suggestion to strengthen the discussion according to the mentioned points effectively improved the quality of the manuscript, and we appreciate this valuable input.

Regarding your inquiry about the potential negative impact of androsterility sources on beverage quality, it holds great significance, given the natural evolution of the market. This aspect corresponds to a second stage of our study, as well as other points you have pointed out. It is worth noting that existing experiences with the inclusion of Ethiopian germplasm have shown favorable results, and in no case have they indicated any negative effect on beverage quality.

---

## [Editor Report · Decision Letter 1]

25 Aug 2023

Identification of sources of male sterility in the Colombian Coffee Collection for the genetic improvement of Coffea arabica L.

PONE-D-23-12131R1

Dear Dr. Arias,

We’re pleased to inform you that your manuscript has been judged scientifically suitable for publication and will be formally accepted for publication once it meets all outstanding technical requirements.

Kind regards,

Abhishek Bohra

Academic Editor

PLOS ONE
---

## [Editor Report · Acceptance letter]

30 Aug 2023

PONE-D-23-12131R1 

Identification of sources of male sterility in the Colombian Coffee Collection for the genetic improvement of *Coffea arabica* L. 

Dear Dr. Suárez:

I'm pleased to inform you that your manuscript has been deemed suitable for publication in PLOS ONE. Congratulations! Your manuscript is now with our production department. 

Kind regards, 

on behalf of

Dr. Abhishek Bohra 

Academic Editor

PLOS ONE